# Characterization and Advancement of an Evaluation Method for the Treatment of Spontaneous Osteoarthritis in STR/ort Mice: GRGDS Peptides as a Potential Treatment for Osteoarthritis

**DOI:** 10.3390/biomedicines11041111

**Published:** 2023-04-06

**Authors:** Mei-Feng Chen, Chih-Chien Hu, Yung-Heng Hsu, Yu-Tien Chiu, Kai-Lin Chen, Steve W. N. Ueng, Yuhan Chang

**Affiliations:** 1Bone and Joint Research Center, Chang Gung Memorial Hospital, Taoyuan 33305, Taiwan; 2Department of Orthopedic Surgery, Chang Gung Memorial Hospital, Taoyuan 33305, Taiwan; 3College of Medicine, Chang Gung University, Taoyuan 33302, Taiwan

**Keywords:** osteoarthritis, STR/ort, cartilage, matrix metallopeptidase-13, SRY-box transcription factor 91

## Abstract

STR/ort mice spontaneously exhibit the typical osteoarthritis (OA) phenotype. However, studies describing the relationship between cartilage histology, epiphyseal trabecular bone, and age are lacking. We aimed to evaluate the typical OA markers and quantify the subchondral bone trabecular parameters in STR/ort male mice at different weeks of age. We then developed an evaluation model for OA treatment. We graded the knee cartilage damage using the Osteoarthritis Research Society International (OARSI) score in STR/ort male mice with or without GRGDS treatment. We measured the levels of typical OA markers, including aggrecan fragments, matrix metallopeptidase-13 (MMP-13), collagen type X alpha 1 chain (COL10A1), and SRY-box transcription factor 9 (Sox9), and quantified epiphyseal trabecular parameters. Compared to the young age group, elderly mice showed an increased OARSI score, decreased chondrocyte columns of the growth plate, elevated expression of OA markers (aggrecan fragments, MMP13, and COL10A1), and decreased expression of Sox9 at the articular cartilage region in elderly STR/ort mice. Aging also significantly enhanced the subchondral bone remodeling and microstructure change in the tibial plateau. Moreover, GRGDS treatment mitigated these subchondral abnormalities. Our study presents suitable evaluation methods to characterize and measure the efficacy of cartilage damage treatments in STR/ort mice with spontaneous OA.

## 1. Introduction

Osteoarthritis (OA) affects 10–15% of older adults (>60 years old) worldwide. It is a degenerative joint disease caused by the wear or breakdown of the cartilage covering the ends of bone joints due to chronic inflammation, leading to localized pain and limited joint movement [1,2]. In healthy individuals, chondrocytes in articular cartilage maintain a dynamic balance between the generation and degradation of the extracellular matrix (ECM), including collagen, proteoglycan, and aggrecan [3,4]. In the early progression of OA, disruption of the ECM balance leads to the upregulation of anabolism and catabolism within the joint until the catabolic rate gradually exceeds the anabolic rate [1]. This imbalance induces inflammation and increases the number of reactive oxygen species. Furthermore, inflammation results in decreased ECM synthesis and enhanced ECM degradation. During OA progression, inflammation stimulates several pathways involved in the release of matrix metalloproteinase (MMP) from chondrocytes, which are not only involved in the upregulation of cartilage-degrading proteases but also in the production of pain in OA cartilage. The best-known collagenases in OA are MMP-1 and -13, which have been demonstrated to promote cartilage breakdown and repress the expression of major ECM molecules, such as aggrecan and collagen [5,6]. Immunostaining of collagen type X alpha 1 chain (COL10A1) and MMP-13 was used as markers of the hypertrophic transformation in articular cartilage [7]. SRY-box transcription factor 9 (Sox9) is an essential transcription factor for chondrogenesis, which is a suppressor of metalloproteinases-induced cartilage degeneration at the early stage of human OA [8]. Thus, the elevated expression of aggrecan fragments, MMP-13, COL10A1, and decreased expression of Sox9 represent typical OA markers.

The treatment of OA aims to reduce pain, prevent ECM degradation, maintain or improve joint function, and minimize disability. Animal models are a critical method for studying OA, as they provide a potential efficacy testing platform for the development of new treatments [9,10,11]. In addition, the STR/1N strain of mice was originally isolated by Strong (1951) in an extensive selective breeding program aimed at identifying tumor-induced resistance characteristics at sites of carcinogen injection [12]. The STR/1N strain was incidentally generated by tandem hybridization between CBA, N, J, and K strains and further selection with another carcinogen (4-methylcholestene) [13]. STR/1N strains exhibit obesity and spontaneous OA at a young age [14]. After some breeding and development by the Orthopaedic Research Institute in Stanmore in the UK, the strain was renamed STR/ort as it is now known [15]. While the genetics of OA susceptibility in STR/ort mice is uncertain, these mice spontaneously develop OA in the knee joint and temporomandibular joints [16,17]. STR/ort mice spontaneously exhibit the OA phenotype at a young age and reveal numerous OA characteristics as those of clinical cases, including disturbed and lost articular cartilage, digested ECM, decreased proteoglycan, and osteophyte formation [14,18]. Therefore, the STR/ort mouse model has been a suitable high-incidence model of spontaneous OA in many studies over the past 20 years [16]; however, there is no detailed data to describe the relationship between cartilage histology, typical OA markers, epiphyseal trabecular bone, and age.

We aimed to evaluate the histological features, OARSI scores, typical OA markers, and subchondral bone plate thickness and quantify the epiphyseal trabecular parameters from 6 to 40 weeks old STR/ort mice. We believe this data can be of great value for verifying new OA therapeutic treatments. On the other hand, since the regulation of integrin function seems to impact cell–matrix and cell–cell interactions in cartilage development, homeostasis, and degradation [19], a priori, chondrocytes can potentially use integrins α1β1, α2β1, α3β1, α5β1, α6β1, and αVβ3 to bind ECM ligands [20]. The soluble pentapeptide Gly-Arg-Gly-Asp-Ser (GRGDS) can act as an integrin antagonism by the soluble integrin recognition [21]; thus, the GRGDS peptides were used as an example of a drug screening model for OA treatment in STR/ort mice.

## 2. Materials and Methods

### 2.1. Experimental Animal Studies

All animal procedures complied with the National Institutes of Health in the United States guidelines and were reviewed and approved by the local Hospital Animal Care and Use Committee (Institutional Animal Care and Use Committee approval number 2018060702, approval date: 25 June 2021). In addition, the Animal Research: Reporting In Vivo Experiments (ARRIVE) guidelines for animal research and submission of studies were followed [22]. STR/ort mice acquired from RIKEN BioResource Research Center (Tsukuba, Japan) were maintained in environmentally controlled rooms (controlled room temperature: 25 ± 1 °C) and subjected to 12-h light–dark cycles. A total of 33 male STR/ort mice were randomly divided into 6 groups of 4–10 mice each for this study. In Figure 1, Figure 2 and Figure 3 and Appendix A, each group represents 4 mice at 6 weeks old, 4 mice at 10 weeks old, 6 mice at 14 weeks old, 10 mice at 22 weeks old, and 4 mice at 40 weeks old. In Figure 4 and Appendix A, STR/ort mice were administered tail vein injections of 0.3 mg/mouse GRGDS domain (G4391, Sigma-Aldrich, St. Louis, MO, USA) (*n* = 5 mice) or vehicle PBS solution (*n* = 4 mice). Mice were anesthetized using an intraperitoneal injection of a 0.01 mL/kg mixture (1:1 *v*/*v*) of tiletamine-zolazepam (Zoletil^®^ 50, Carros, France) and xylazine hydrochloride (Rompun^®^ Bayer HealthCare, Berlin, Germany) and sacrificed. The tibiofemoral joint was immediately fixated in 10% formaldehyde and subjected to microcomputed tomography (micro-CT) analysis and safranin O and immunofluorescence staining. Each protocol was carried out by an investigator blinded to the experimental conditions.

### 2.2. Safranin O and Immunofluorescence Staining

The tibiofemoral joint samples were harvested in neutral buffered formalin (10%), incubated in a rapid decalcifier solution, trimmed, and embedded in paraffin. Subsequently, 5 μm thick sections were stained with (1) safranin O, (2) aggrecan fragments (1:100, AF1220, R&D system, Inc., Minneapolis, MN, USA), (3) Sox9 (1:50, sc-166505, Santa Cruz Biotechnology, Inc., Delaware, CA, USA), and (4) MMP13 (1:100, GTX100665, GeneTex, Irvine, CA, USA). For Safranin O staining, sections were incubated in Weigert’s iron hematoxylin (HT1079, Sigma-Aldrich, St. Louis, MO, USA), 0.08% fast green (F7252, Sigma-Aldrich, St. Louis, MO, USA), and 0.1% Safranin O solution (S2255, Sigma-Aldrich, St. Louis, MO, USA). Primary antibodies against aggrecan fragments, Sox9, and MMP13 were used for immunofluorescence (IF) staining. Samples were subsequently incubated with a secondary antibody, namely Alexa Fluor 488-conjugated anti-goat IgG (1:500, SA5-10086, Invitrogen, Carlsbad, CA, USA), Alexa Fluor 488-conjugated anti-mouse IgG (1:500, R37114, Invitrogen, Carlsbad, CA, USA), and Alexa Fluor 488-conjugated anti-rabbit IgG (1:500, A21206, Invitrogen, Carlsbad, CA, USA), for 60 min at 25 °C. DAPI (4′, 6-diamidino-2-phenylindole) staining was used for nuclear staining (1:1000, D1306; Invitrogen; Waltham, MA, USA). Safranin-O-stained slides were digitalized using a NanoZoomer S360 digital slide scanner (Hamamatsu Photonics, Hamamatsu, Japan). Each image was acquired under a microscope (DFC7000 T; Leica Microsystems, Wetzlar, Germany). IF slides were digitized using a TissueFAXS System (TissueGnostics^®^, Vienna, Austria) coupled onto a Zeiss Axio Imager Z1 microscope (Zeiss, Jena, Germany). Image acquisition was performed using the TissueFAXS System (TissueGnostics^®^, Vienna, Austria). The signal of IF occurs around the area of articular cartilage. We used TissueQuest software version 7.0 (TissueGnostics^®^, Vienna, Austria) to quantify the fluorescent signal.

### 2.3. OARSI Score

Coronal (frontal) histological sections were stained with Safranin-O, and cartilage damage was scored by two blinded observers using the OARSI scoring system by Glasson et al. [23]. Both right and left knee joints were serial coronal sections (5 μm) and stained with Safranin-O. The histological sections were examined by blinded observers using scoring cartilage pathology at all four quadrants compartments (medial femoral condyle, lateral femoral condyle, medial tibial plateau, lateral tibial plateau).

### 2.4. Micro-CT Bone Imaging

Nondestructive ultrastructural bone analysis was performed with a SkyScan 1176 micro-CT scanner (Bruker, Kontich, Belgium). Samples were wrapped in saline-soaked gauze and subsequently scanned with the following parameters: the pixel size was 9 μm, the voltage was 60 kVp, the current was 417 μA, the filter was 0.5 mm aluminum, and the exposure time was 1000 ms. The cross-sectional images were reconstructed using GUP-NRecon software (version 1.7.4.6, Skyscan, Kontich, Belgium) and analyzed with the Skyscan CTAn program (version 1.20.8.0, Bruker, Kontich, Belgium). The area of articular cartilage was further analyzed. We distinguished specific regions in the subchondral bone, which is epiphysis. Epiphyseal trabeculae from the subchondral bone plate were automatically isolated using CTAn software. Subchondral bone plate thickness (PI.Th.), trabecular parameters (Tb.Th, Tb.N, and Tb.Sp), bone volume (BV/TV), and total porosity [Po(tot)] in the medial- and lateral-tibial plateau were calculated using CTAn software. Illustrative images were obtained with the Skyscan Dataviewer program (version 1.6.0.0, Bruker, Kontich, Belgium).

### 2.5. Statistical Analysis

Quantitative data are presented as means ± standard errors of the mean. Results indicated in Figure 1B,C, Figure 2B,C, and Figure 3C, and Appendix A were analyzed using one-way analyses of variance (ANOVA) followed by Tukey’s post hoc tests. The observations shown in Figure 1D were analyzed using simple linear regression. Supplementary Figure 4 was analyzed using two-way ANOVA followed by Šídák’s multiple comparisons tests. The results presented in Figure 4C–E,G,H and Appendix A were analyzed using an unpaired *t*-test. Figure 4J and Appendix A were analyzed using two-way ANOVA followed by Šídák’s multiple comparisons tests. Statistical calculations were performed using GraphPad Prism 9.3.1 (GraphPad Inc., San Diego, CA, USA). Two-tailed *p*-values < 0.05 were considered statistically significant.

## 3. Results

### 3.1. Histological Assessment of OA Severity Associated Cartilage Damage with Age

Patients with clinically diagnosed OA present a continuously progressive disease state with partially but incomplete cartilage degeneration and chondrocyte loss. Thus, we carefully surveyed the dynamic and spontaneous changes of cartilage damage in STR/ort mice with increasing age. We analyzed the Safranin-O-stained image and measured the Osteoarthritis Research Society International (OARSI) Score of male knee joints from the 6 to 40 weeks old mice (Figure 1A–C). A smooth cartilage surface and normal morphology of chondrocyte nucleus were observed in 6- and 10-week-old mice (Figure 1A). The fibrillar cartilage was observed in 14-week-old mice. The cartilage surface showed vertical clefts and decreased the number of chondrocytes observed in 22-week-old mice, which is similar to human OA symptoms. The OARSI score was significantly increased with age in the medial tibial plateau (MTP), lateral tibial plateau (LTP), medial femoral condyle (MFC), and lateral femoral condyle (LFC) (Figure 1B and Appendix A). The matrix-nonproducing chondrocytes were significantly increased at the region of calcium cartilage in the MTP but not in the LTP; moreover, cartilage morphology was significantly lost in 40-week-old mice, with a decreased and disorganization of chondrocyte columns in the growth plate with age (Figure 1D,E and Appendix A).

### 3.2. Assessment of Articular Chondrocyte Hypertrophy Associated Markers with Age

To further investigate the effect of age on the articular chondrocyte hypertrophy-related molecules in STR/ort knee, we performed IF staining for the matrix degradation marker with aggrecan fragments and metalloproteinase with MMP13, and for the chondrogenic transcription factor with Sox9 (Figure 2A). The entire knee joint split out for MTP, MFC, LTP, and LFC to quantify the fluorescence signal in the articular cartilage region. The results of the fluorescence signal indicated elevated expression of aggrecan fragments and MMP13, decreased expression of Sox9, and no changed expression of COL10A1 at the region of MTP and LTP (Figure 2B), which signifies matrix degradation that typifies OA progression. Additionally, it also detected significant changes in these typified OA markers at the region of MFC and LFC (Appendix A).

### 3.3. Subchondral Bone Plate Thickness and Epiphyseal Trabecular Parameters

Micro-CT three-dimensional reconstructions of the knee joints of STR/ort mice aged 6–40 weeks old showed that the 14 to 40-week-old groups exhibited increased subchondral bone mineralization, particularly at the medial side of the tibial and femur (Figure 3A, coronal section). The CT images also showed significantly uneven trabecular bone distribution in the tibia (Figure 3A, horizontal section). The epiphysis was manually selected as representative of subchondral bone from the MTP and LTP, and the epiphyseal trabeculae were separated from the subchondral bone plate, and trabecular bone morphometric parameters were analyzed (Figure 3B).

The quantified subchondral region by μCT analysis indicated that, compared to 6-week-old mice, 10–40-week old mice showed a significant increase in subchondral bone plate thickness (PI.Th), subchondral trabecular parameters (Tb.Th and Tb.Sp), and subchondral bone volume density (BV/TV), particularly at the medial side in tibial (Figure 3C). In addition, the subchondral bone total porosity [Po(tot)] increased with age. These OA-related subchondral abnormalities worsen with age (Figure 3A–C), and the parameters were significantly changed with aging on the medial side compared to the lateral side (Appendix A). We demonstrated that aging STR/ort significantly changes subchondral bone parameters, particularly on the medial side, including PI.Th, Tb.Sp, Tb.N, BV/TV, and Po(tot).

### 3.4. GRGDS Treatment Prevents Cartilage Degeneration and Subchondral Bone Mineralization

We further determined the efficacy of GRGDS treatment in OA progression, used as an example to develop and validate a drug screening model for OA treatment in STR/ort mice (Figure 4). Compared to the vehicle control group, GRGDS-treated mice demonstrated reduced cartilage degeneration, decreased OARSI score at the MTP, decreased matrix-nonproducing chondrocyte in the calcium cartilage of MTP and LTP, and prevented disorganization of chondrocyte columns in the growth plate (Figure 4A–E). However, the OARSI scores were unchanged at the LTP, MFC, and LFC regions; the matrix-nonproducing chondrocyte was unchanged at the non-calcium cartilage regions (Appendix A).

To further investigate the effect of GRGDS treatment on the typified OA markers in cartilage, we performed IF staining for the aggrecan fragments, MMP13, Sox9, and COL10A1 (Figure 4F). Upon quantification of the staining in the MTP region, we detected the inhibited expression of aggrecan fragments and MMP13, enhanced expression of Sox9, and no changed expression of COL10A1 in GRGDS-treated STR/ort mice (Figure 4G). In the LTP region, we confirmed the inhibited expression of MMP13, enhanced expression of Sox9, and no changed expression of aggrecan fragments and COL10A1 in GRGDS-treated STR/ort mice (Figure 4H). Furthermore, GRGDS treatment also inhibited the expression of aggrecan fragments and MMP13 in the regions of MFC and LFC (Appendix A). Finally, three-dimensional micro-CT analysis was used to measure the effects of GRGDS treatment on joint mineralization and subchondral bone remodeling. According to the results of CT images, compared to vehicle control mice, the GRGDS-treated group showed attenuated subchondral bone sclerosis, especially at the medial knee side (Figure 4I). As compared to vehicle control mice, GRGDS-treated STR/ort mice showed lower subchondral trabecular thickness (Tb.Th), lower subchondral bone volume density (BV/TV), and higher subchondral bone porosity (Figure 4J). These OA-associated subchondral abnormalities were alleviated by GRGDS treatment. These effects might be secondary to improved OA or due to the regulation of hypertrophic chondrocyte biology in the subchondral bone. Nevertheless, GRGDS treatment had no effect on the subchondral bone plate thickness (PI.Th), subchondral trabecular number (Tb.N), and subchondral trabecular spacing (Tb.Sp) (Appendix A).

## 4. Discussion

The severity of articular cartilage loss is a key indicator of OA progression. The male STR/ort mice develop knee OA spontaneously and, thus, serve as an excellent animal model for OA studies [24]. Although many studies describe the anatomical and clinical features of the knee joint of STR/ort OA mice as similar to those of OA patients, cartilage histological score analyses of STR/ort mice present uncertain results [14,16,25]. The cartilage thickness is approximately 30 μm, 300 μm, and 2.3 mm in mice, rats, and humans, respectively [26,27]. This is considered to be a challenge for assessment using the OARSI score due to the thin cartilage layers. Thus, this study is dedicated to a detailed evaluation of the degree of cartilage damage using OARSI scores in STR/ort mice aged 14 to 40 weeks old. Additionally, the GRGDS peptides were also used as an example of a drug screening model for OA treatment in STR/ort mice.

To our knowledge, this study is the first to provide data to measure the expression level of aggrecan fragments and MMP-13 in STR/ort mice aged 6 to 22 weeks old. Aggrecan fragments (G1-IGD-G2 domain) and MMP-13 are markers for chondrocyte hypertrophy-like changes and cartilage degradation in OA [5,28]. Aggrecan has a core protein to link sulfated glycosaminoglycan (GAG) chains covalently and forms supramolecular complexes with hyaluronan [29]. Under normal physiological conditions, aggrecan does not exist alone in the extracellular matrix but in the form of proteoglycan aggregates [29]. Aggrecan plays a key role in bearing compressive loads in articular cartilage, and aggrecan fragments are released into the synovial fluid under osteoarthritis conditions [30,31,32]. Moreover, the aggrecan G1-IGD-G2 domain is the site of matrix metalloproteinases (MMP) attack on aggrecan during pathological cartilage degradation; G1-IGD-G2 appears to be involved in the physiological turnover of aggrecan [33]. On the other hand, MMP-13, also named collagenase-3, a member of the MMP family of neutral endopeptidases, is highly overexpressed in hypertrophic chondrocytes and synovial cells in osteoarthritis [34].

The SOX9 is a suppressor of metalloproteinases-induced cartilage degeneration at the early stage of human osteoarthritis [8]. When inactive, Sox9, an essential gene in the physiological control of cartilaginous tissues in adult mice, causes growth plate shrinkage and proteoglycan loss in the articular cartilage [35]. Sox9 transcription factor is essential for chondrogenesis and is down-regulated in OA [36]. In addition, it is known that integrins are heterodimeric transmembrane receptors that mediate multiple biological functions and play a key role in the pathogenesis of osteoarthritis, which may provide new targets for the development of OA therapy [19]. GRGDS peptides are known to activate alphaVbeta3; moreover, integrin alphaVbeta3 signaling protects chondrocytes and represents a therapeutic target for therapies to prevent cartilage degeneration in OA [37]. Many studies have used RGD or RGDS as biomaterials to promote chondrocyte attachment and proliferation [38,39,40]. Therefore, we evaluated the therapeutic effect of GRGDS on OA.

In this study, we first provided data to demonstrate that GRGDS domain treatment prevents the decrease in Sox9 expression caused by aging and also maintains the chondrocyte column structure of growth plates in aged STR/ort mice. In addition, many reports have shown that it either alleviates inflammation or enhances SOX9 and could be used as a potential therapeutic agent for human OA. Overexpression in enthesis promotes mineralization by inflammation stimuli in ankylosing spondylitis [41]. The long intergenic non-coding RNA, a regulator of reprogramming (linc-ROR), promotes mesenchymal stem cells’ chondrogenesis and cartilage formation via regulating Sox9 expression [42]. The upregulation of lncRNA H19 attenuates inflammation and ameliorates cartilage damage and chondrocyte apoptosis in OA by upregulating tumor protein p53 (TP53), IL-38, and by activating IL-36R [43]. A recent study also demonstrates that Sox9 keeps growth plates and articular cartilage healthy by inhibiting chondrocyte dedifferentiation/osteoblastic redifferentiation [36]. Sox9 also regulates ribosome activity and subsequent cartilage extracellular matrix production of chondroprogenitors in the growth plate in vivo [44]. The delivery of therapeutic recombinant adenovirus-mediated Sox9 gene transfer to sites of osteochondral defects repairs the articular cartilage and reduces perifocal osteoarthritic changes in OA animals [45]. Since we demonstrate that GRGDS treatment prevents the decrease in Sox9 expression, we believe that our findings can facilitate the development of novel strategies for the treatment of OA, contributing to the field of chondrocyte biology.

Finally, knee OA is characterized by increased subchondral bone plate thickness and trabecular bone volume. The subchondral trabecular bone remodeling and microstructure change in the tibial plateau is associated with hip–knee–ankle angle and cartilage degradation, thus, resulting in accelerated knee OA progression [46]. Although subchondral bone changes seem to be secondary in late-stage OA of the knee caused due to cartilage damage, it is an objective, quantifiable indicator. This study demonstrates that aging significantly changes subchondral bone parameters, particularly on the medial side, including PI.Th, Tb.Sp, Tb.N, BV/TV, and Po(tot) in STR/ort mice. Moreover, the chondrocyte loss associated with subchondral abnormalities is mitigated by GRGDS treatment.

## Figures and Tables

**Figure 1 biomedicines-11-01111-f001:**
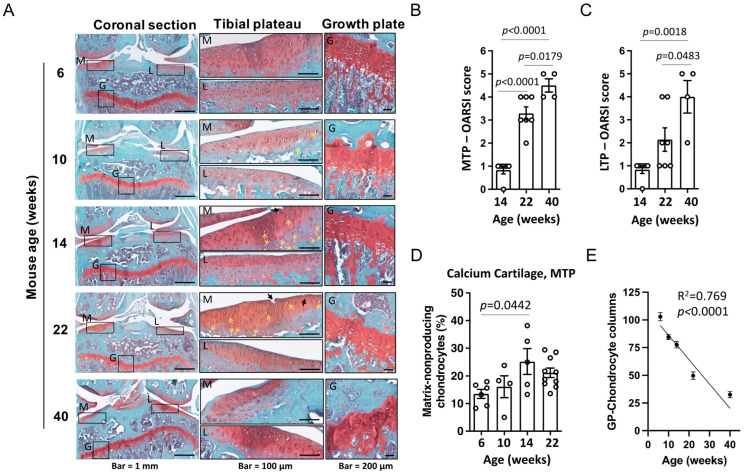
The articular cartilage of STR/ort mice showing varying stages of histological OA with age. (**A**) The articular cartilage changes occurred in the tibial plateau and femoral condyle. Magnified images of regions marked by black boxes in A. Normal cartilage is present with a smooth surface at 6 and 10 weeks of age. Early OA with small fibrillations occurred in the medial tibial plateau at 14 weeks of age. Advancing OA with vertical clefts down to the layer immediately below the superficial layer and some loss of surface lamina is shown at 22 weeks of age. Severe OA with loss of Safranin-O staining and loss of articular cartilage down to the subchondral bone is shown at 40 weeks old. Arrows in A indicate representative vertical clefts (black arrows) or matrix-nonproducing (yellow arrows) chondrocytes. An altered columnar structure and disorganization of chondrocyte columns in growth plates result with age. (**B**,**C**) The OARSI scoring system showed that 22-week-old mice were more severely injured than 14-week-old mice at the MTP, and 40-week-old mice were more severely injured than 22-week-old mice at the LTP. (**D**) The matrix-nonproducing chondrocytes were significantly increased at the region of calcium cartilage in the MTP. (**E**) The decreased and disorganization of chondrocyte columns in the growth plate with age. Abbreviations: M, medial side; L, lateral side; G, growth plate; MTP, medial tibial plateau; LTP, lateral tibial plateau.

**Figure 2 biomedicines-11-01111-f002:**
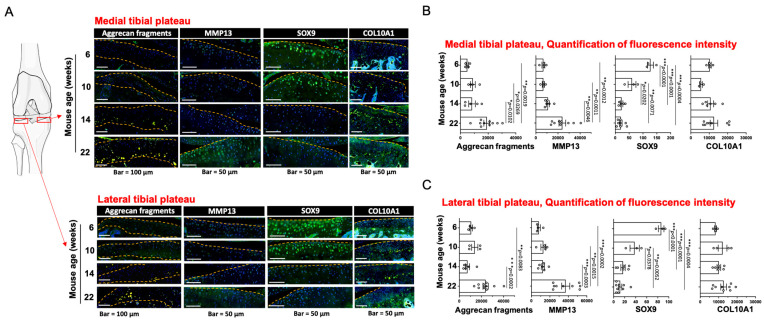
Changed expression level of OA markers associated with age is detected in articular cartilage. (**A**) Representative images of aggrecan fragments, aggrecan fragments, MMP-13, Sox9, and COL10A1 IF staining (green) in whole articular cartilage from STR/ort mice with indicated weeks old. DAPI (blue) stains nuclei; yellow dashed lines define the cartilage region. (**B**) Quantification of fluorescence in the region of whole articular cartilage in MTP. (**C**) Quantification of fluorescence in the region of whole articular cartilage in LTP. Abbreviations: MTP, medial tibial plateau; LTP, lateral tibial plateau; Sox9, SRY-box transcription factor 9; MMP-13, Matrix metalloproteinase-13; COL10A1, collagen type X alpha 1 chain.

**Figure 3 biomedicines-11-01111-f003:**
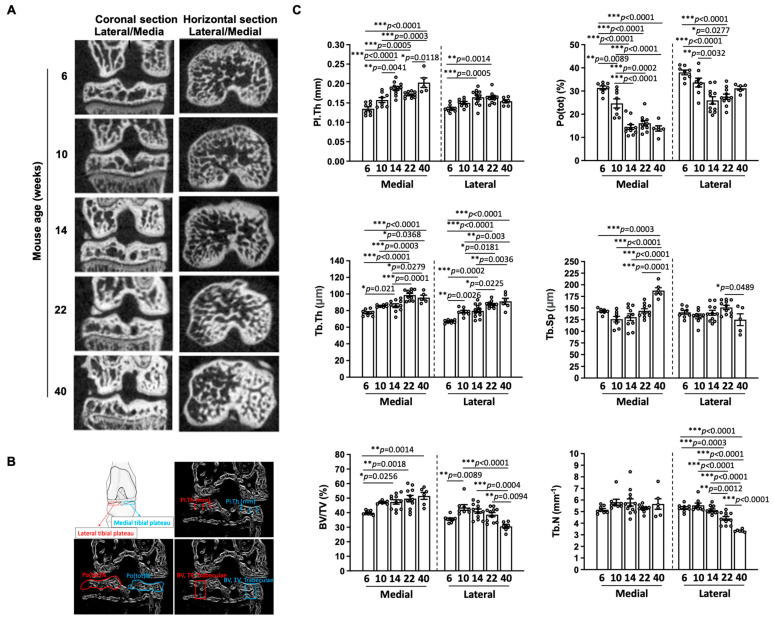
Subchondral bone plate thickness and epiphyseal trabecular bone increase with age in STR/ort joints. (**A**) Representative cross and horizontal sections of knee joints from STR/ort mice. The 3D reconstruction of a proximal tibia with a plane indicating the location from which the cross and horizontal sections were taken. Subchondral bone sclerosis was found in the medial region of the epiphysis, observed from 14 to 40 weeks old. (**B**) Representative the selected regions of tibial epiphysis for 3D-morphometric analysis. The medial and lateral subchondral bone plate and underlying epiphyseal trabecular bone were analyzed separately. Epiphysis was manually selected as representative of subchondral bone. The epiphyseal trabeculae were split from the subchondral bone plate, and trabecular bone morphometric parameters were calculated. (**C**) Trabecular bone morphometric parameters were calculated using micro-CT analysis. A subset of the medial and lateral subchondral bone plate and underlying epiphyseal trabecular bone were analyzed separately, as indicated for the medial and lateral side. Subchondral bone plate thickness (PI.Th), trabecular parameters (Tb.Th and Tb.Sp), and bone volume (BV/TV) were increased with age in STR/ort mice. Subchondral bone total porosity [Po(tot)] decreased with age. Abbreviations: BV, bone volume; TV, tissue volume; Tb.Th, trabecular thickness; Tb.N, trabecular number; Tb.Sp, trabecular spacing.

**Figure 4 biomedicines-11-01111-f004:**
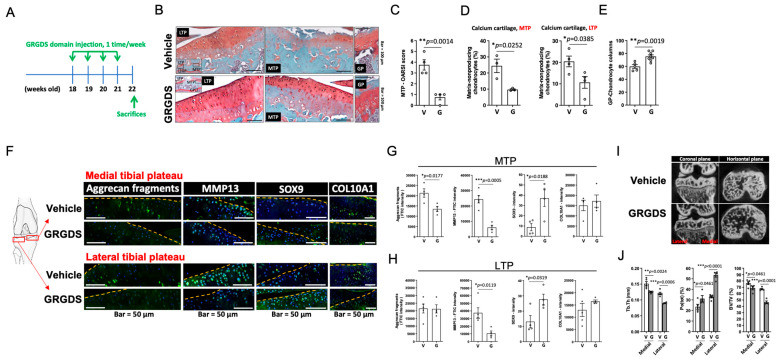
GRGDS domain prevents OA progression in STR/ort mice. (**A**) Schematic representation of the timeline of drug treatment. STR/ort mice were tail vein injected once a week with vehicle or GRGDS for a total of four consecutive weeks. Mice were sacrificed at 22 weeks old. (**B**) Representative images are Safranin-O stained articular cartilage and growth plate. Advancing OA with severe fibrillations and loss of articular cartilage occurred in the medial tibial plateau at 22 weeks of age in the vehicle control groups. The GRGDS domain prevents the loss of articular cartilage in mice. (**C**) GRGDS domain treatments decreased the OARSI score in the MTP. (**D**) GRGDS inhibits the matrix-nonproducing chondrocytes increased in calcium cartilage. (**E**) GRGDS maintain the numbers of chondrocyte columns in the growth plate. (**F**) Representative images of aggrecan fragments, MMP-13, Sox9, and COL10A1 IF staining (green) in whole articular cartilage from STR/ort mice with indicated weeks old. DAPI (blue) stains nuclei; yellow dashed lines define the cartilage region. (**G**,**H**) Quantification of fluorescence in the region of MTP and LTP. (**I**) The 3D reconstruction of a proximal tibia with a plane indicating the location from which the sections were taken. Subchondral sclerosis in the medial region of the epiphysis was observed in vehicle control group, whereas the GRGDS treatment prevented subchondral sclerosis. (**J**) Epiphyseal trabecular bone parameters were calculated using micro-CT analysis. A subset of the medial and lateral subchondral bone plate and underlying epiphyseal trabecular bone were analyzed separately, as indicated for the medial and lateral side. GRGDS domain treatment prevented Tb.Th and BV/TV increases in 22-week-old STR/ort mice and restored Po(tot) values. Abbreviations: MTP, medial tibial plateau; LTP, lateral tibial plateau; GP, growth plate; Sox9, SRY-box transcription factor 9; MMP-13, Matrix metalloproteinase-13; COL10A1, collagen type X alpha 1 chain; BV, bone volume; TV, tissue volume; Tb.Th, trabecular thickness; Po(tot), subchondral bone total porosity.

## Data Availability

The data presented in this study are available on request from the corresponding author.

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
