# Peer review of "Characterization and Advancement of an Evaluation Method for the Treatment of Spontaneous Osteoarthritis in STR/ort Mice: GRGDS Peptides as a Potential Treatment for Osteoarthritis"

_biomedicines, 2023, doi:10.3390/biomedicines11041111_

Round 1

Reviewer 1 Report

An article presents interesting observations which can potentially worthy in future examination. Methodology, statistics analysis, and form of presentation all are prepared very well. I recommend to publish an article Characterization and advancement of an evaluation method for the treatment of spontaneous osteoarthritis in STR/ort mice in Biomedicines.

Author Response

Response to Reviewer 1 Comments

Point: An article presents interesting observations which can potentially worthy in future examination. Methodology, statistics analysis, and form of presentation all are prepared very well. I recommend to publish an article Characterization and advancement of an evaluation method for the treatment of spontaneous osteoarthritis in STR/ort mice in Biomedicines.

Response: We appreciate the reviewer’s comment.

Reviewer 2 Report

Osteoarthritis is a serious problem which fully justifies the research undertaken. The research model was chosen correctly, with the caveat that the biomechanics of the mouse and human knee joint are different.  The subchondral bone, along with the joint synovium, is crucial for the proper functioning of the articular cartilage. The study groups were sufficiently large to draw meaningful conclusions. The research methodology is appropriate and the laboratory equipment is of high quality.
Unfortunately, the figures are too small and make it impossible to analyse the histological images obtained (Figure 3 is legible). I would ask for enlargement to improve the readability of the images.
The literature is sufficient. The discussion is adequate and the conclusions are supported by the results.

Author Response

Response to Reviewer 2 Comments

Point: Osteoarthritis is a serious problem which fully justifies the research undertaken. The research model was chosen correctly, with the caveat that the biomechanics of the mouse and human knee joint are different.  The subchondral bone, along with the joint synovium, is crucial for the proper functioning of the articular cartilage. The study groups were sufficiently large to draw meaningful conclusions. The research methodology is appropriate and the laboratory equipment is of high quality.

Unfortunately, the figures are too small and make it impossible to analyse the histological images obtained (Figure 3 is legible). I would ask for enlargement to improve the readability of the images.

The literature is sufficient. The discussion is adequate and the conclusions are supported by the results.

Response: We appreciate the reviewer’s comment. Our improvement method consists of two parts. First, we have enlarged the figures; please refer to the revised version. Second, we will provide high-resolution images for journal publication to avoid low image resolution due to the size limit of attachments in the upload submission system.

Reviewer 3 Report

This manuscript develops evaluation methods of osteoarthritis in STR/ort mice. The different symptoms showed using images and graphs. However, all methods used in this study are commonly used in osteoarthritis research. Therefore, novelty is limited. 

Comments

Figures and graphs are tiny with low resolution; it is difficult to check.  

In fig 4. (B), the MTP tissue picture of the GRGDS group is unsuitable for comparison because the center area is damaged. 

Author Response

Response to Reviewer 3 Comments

Point 1: This manuscript develops evaluation methods of osteoarthritis in STR/ort mice. The different symptoms showed using images and graphs. However, all methods used in this study are commonly used in osteoarthritis research. Therefore, novelty is limited. 

Comments: Figures and graphs are tiny with low resolution; it is difficult to check.  

Response 1: We appreciate the reviewer’s comment. Our improvement method consists of two parts. First, we have enlarged the figures; please refer to the revised version. Second, we will provide high-resolution images for journal publication to avoid low image resolution due to the size limit of attachments in the upload submission system.

Point 2: In fig 4. (B), the MTP tissue picture of the GRGDS group is unsuitable for comparison because the center area is damaged. 

Response 2: We appreciate the reviewer’s comment. For the GRGDS treated group in Fig 4B, we have changed to a more suitable image; please see the revised Figure 4 in lines 308. An updated version of Fig 4b is also presented below.

Reviewer 4 Report

This is a very interesting and focused paper on OA, a worldwide painful degenerative disease. There is a good introduction and the M&M are correct, including statistical analysis. As a suggestion, concerning OA biomarkers, it has been proposed that Pit-1 increases the mARN expression and the activity of MMP-1 and MMP-13 (authors could look for references about it). As authors use MMP-13 but not MMP-1, the introduction of Pit-1 biomarker could increase the number of tools for future studies.

About results, they are clear, concise and well structured. Figures are brilliant, so clear in relation to the complexity of the studied biological system and the figure legends are very helpful and comprehensive. For a more friendly format, I suggest the insertion of Figures in throughout the text close to the appropriate paragraph instead of all figures at the end of results. Some of the supplemental Figures could be incorporated to the regular paper, but this is an author´s decision.  

Concerning discussion, it is demonstrated that the treatment with the GRGDS pentapeptide prevents the decrease in Sox9 expression, and supposedly this ameliorates the loss of chondrocytes associated with subchondral abnormalities. This observed effect could be helpful for the development of novel strategies leading to the OA treatment. However, discussion about the rationality of the use of that peptide is missing. As far as I know, the RGD motif is well known, but the extension to the pentapeptide is not so. I have several questions concerning this important point. Are there any references about the use of that peptide? Do the glycine and serine residues placed at the N- and C-terminal side of the motif improve the effect in relation to the RGD tripeptide? Is the pentapeptide an invention of the authors? Please, add some brief comments or appropriate references on this regard.  

Related to that, there is a paragraph at the end of the introduction related to the potential use of a series of integrins (α1β1, α2β1, α3β1, α5β1, α6β1, and αVβ3) to bind ECM ligands. However, this important point is not further mentioned at the discussion. I suggest changing the location of that paragraph to the discussion section to improve it.

The following minor points should be addressed:

Line 47-48: replace collagenase ……. has by collagenases …… have (two MMP are mentioned)

Line 59: indicate that STR/1N is a mice strain.

Line 107: 2.2. Safranin O and Immunofluorescence Staining. A reference supporting the use of safranin O for tibiofemoral joints and the specificity of this staining would be improve the quality of M&M at line 110. Is that stain complementary to the use of aggrecan fragments?

Author Response

Response to Reviewer 4 Comments

Point 1: This is a very interesting and focused paper on OA, a worldwide painful degenerative disease. There is a good introduction and the M&M are correct, including statistical analysis. As a suggestion, concerning OA biomarkers, it has been proposed that Pit-1 increases the mARN expression and the activity of MMP-1 and MMP-13 (authors could look for references about it). As authors use MMP-13 but not MMP-1, the introduction of Pit-1 biomarker could increase the number of tools for future studies.

Response 1: We appreciate the reviewer’s comments, guidance and suggestions. Although this study involved detecting the expression of MMP13, SOX9, aggrecan fragments, and COL10A1 in the progression of OA, MMP1, and Pit1 are also biomarkers of OA. Our future experiments will measure MMP1 and Pit1.

Point 2: About results, they are clear, concise and well structured. Figures are brilliant, so clear in relation to the complexity of the studied biological system and the figure legends are very helpful and comprehensive. For a more friendly format, I suggest the insertion of Figures in throughout the text close to the appropriate paragraph instead of all figures at the end of results. Some of the supplemental Figures could be incorporated to the regular paper, but this is an author´s decision.  

Response 2: Thank you for the reviewer’s suggestion. We have inserted the individual figures of the data close to the appropriate paragraphs in the Results section; that is, each figure has been placed individually at the appropriate paragraph position. Please refer to the revised version.

Point 3: Concerning discussion, it is demonstrated that the treatment with the GRGDS pentapeptide prevents the decrease in Sox9 expression, and supposedly this ameliorates the loss of chondrocytes associated with subchondral abnormalities. This observed effect could be helpful for the development of novel strategies leading to the OA treatment. However, discussion about the rationality of the use of that peptide is missing. As far as I know, the RGD motif is well known, but the extension to the pentapeptide is not so. I have several questions concerning this important point. Are there any references about the use of that peptide? Do the glycine and serine residues placed at the N- and C-terminal side of the motif improve the effect in relation to the RGD tripeptide? Is the pentapeptide an invention of the authors? Please, add some brief comments or appropriate references on this regard. 

Response 3: We appreciate the reviewer’s comments. The responses to the reviewer's questions about GRGDS are as follows. First, the GRGDS peptide was purchased.

Second, it is known that integrins are heterodimeric transmembrane receptors that mediate multiple biological functions and play a key role in the pathogenesis of osteoarthritis, which may provide new targets for the development of OA therapy1. GRGDS peptides are known to activate alphaVbeta3; moreover, integrin alphaVbeta3 signaling protects chondrocytes and represents a therapeutic target for therapies to prevent cartilage degeneration in OA2. Therefore, we evaluated the therapeutic effect of GRGDS on OA.

Third, many studies use RGD or RGDS as biomaterials to promote chondrocyte attachment and proliferation3-5. Since GRGDS is easy to purchase and obtain, we used GRGDS as a substitute for RGD and RGDS.

References:

  1. Sumsuzzman DM, Khan ZA, Choi J, Hong Y. Assessment of functional roles and therapeutic potential of integrin receptors in osteoarthritis: A systematic review and meta-analysis of preclinical studies. Ageing Res Rev 2022; 81: 101729.
  2. Wang Z, Boyko T, Tran MC, LaRussa M, Bhatia N, Rashidi V, et al. DEL1 protects against chondrocyte apoptosis through integrin binding. J Surg Res 2018; 231: 1-9.
  3. Zhang J, Mujeeb A, Du Y, Lin J, Ge Z. Probing cell-matrix interactions in RGD-decorated macroporous poly (ethylene glycol) hydrogels for 3D chondrocyte culture. Biomed Mater 2015; 10: 035016.
  4. Jung HJ, Park K, Kim JJ, Lee JH, Han KO, Han DK. Effect of RGD-immobilized dual-pore poly(L-lactic acid) scaffolds on chondrocyte proliferation and extracellular matrix production. Artif Organs 2008; 32: 981-989.
  5. Tan H, Huang D, Lao L, Gao C. RGD modified PLGA/gelatin microspheres as microcarriers for chondrocyte delivery. J Biomed Mater Res B Appl Biomater 2009; 91: 228-238.

Point 4: Related to that, there is a paragraph at the end of the introduction related to the potential use of a series of integrins (α1β1, α2β1, α3β1, α5β1, α6β1, and αVβ3) to bind ECM ligands. However, this important point is not further mentioned at the discussion. I suggest changing the location of that paragraph to the discussion section to improve it. 

Response 4: We have edited the relevant parts of the discussion. The revised manuscript reads: “In addition, it is known that integrins are heterodimeric transmembrane receptors that mediate multiple biological functions and play a key role in the pathogenesis of osteoarthritis, which may provide new targets for the development of OA therapy19. GRGDS peptides are known to activate alphaVbeta3; moreover, integrin alphaVbeta3 signaling protects chondrocytes and represents a therapeutic target for therapies to prevent cartilage degeneration in OA37. Many studies use RGD or RGDS as biomaterials to promote chondrocyte attachment and proliferation38-40. Therefore, we evaluated the therapeutic effect of GRGDS on OA.

In this study, we first provided data to demonstrate that GRGDS domain treatment prevents the decrease in Sox9 expression caused by aging, and also maintains the chondrocyte column structure of growth plates in aged STR/ort mice.” (lines 364–375 in the revised manuscript).

Point 5: The following minor points should be addressed: Line 47-48: replace collagenase ……. has by collagenases …… have (two MMP are mentioned)

Response 5: We have edited this sentence. The revised manuscript reads: “The best-known collagenases in OA are matrix metalloproteinase (MMP)-1 and -13,…” (lines 51-52 in the revised manuscript).

Point 6: Line 59: indicate that STR/1N is a mice strain.

Response 6: We have edited this sentence. The revised manuscript reads: “In addition, the STR/1N strain of mice was originally isolated by Strong (1951) in an extensive selective breeding program aimed at identifying tumor-induced resistance characteristics at sites of carcinogen injection12.” (lines 63–65 in the revised manuscript).

Point 7: Line 107: 2.2. Safranin O and Immunofluorescence Staining. A reference supporting the use of safranin O for tibiofemoral joints and the specificity of this staining would be improve the quality of M&M at line 110. Is that stain complementary to the use of aggrecan fragments?

Response 7: We appreciate the reviewer’s comments. The results of safranin O staining and aggrecan fragment staining can complement each other. In addition, the intensity of safranin O staining was directly proportional to the proteoglycan content in cartilage tissue. The staining of aggrecan fragments is specific for degraded aggrecan.

Reviewer 5 Report

Authors proposed a paper entitled “Characterization and advancement of an evaluation method for the treatment of spontaneous osteoarthritis in STR/ort mice” for the publication in Biomedicines, MDPI.

The paper has a good scientific soundness, but I would ask some minor revisions.

I suggest the addition of an abbreviation list, according to the guidelines of this Journal.

Line 47. “and MMP-13(matrix” a space is needed here.

Line 78. “because” I would suggest using expressions beginning with “since”

Line 80. “a priori” should be in italics.

Line 134. “5μm” a space is needed here.

Line 188. “indicate elevated expression” maybe better “indicated”? Moreover, I would re-consider the use of “elevated” and I would consider its substitution.

Line 191. “we also detected” generally I do not advice the use of personal expressions such as “we”.

Figure 1bcd. Figure 1b seems to be characterized by a composition of two sub-figures. Since the other figures are named as b, c, d, I suggest to split figure 1b in figure b, 1c, and then 1d and 1e. Moreover, these figures have not comparable x and y scales. I suggest to use the same scale for axis.

Line 242. It is not necessary to add this line reporting “Figure 1”. Same observation for figure 2 and figure 3, that are even included in the diagrams areas. Please remove them.

Figure 3c are not readable histograms. Please, enlarge them.

Author Response

Response to Reviewer 5 Comments

Point 1: Authors proposed a paper entitled “Characterization and advancement of an evaluation method for the treatment of spontaneous osteoarthritis in STR/ort mice” for the publication in Biomedicines, MDPI. The paper has a good scientific soundness, but I would ask some minor revisions. I suggest the addition of an abbreviation list, according to the guidelines of this Journal. Line 47. “and MMP-13(matrix” a space is needed here.

Response 1: We appreciate the reviewer’s comments. We have edited this sentence. The revised manuscript reads: “The best-known collagenases in OA are matrix metalloproteinase (MMP)-1 and -13,……” (lines 51–52 in the revised manuscript).

Point 2: Line 78. “because” I would suggest using expressions beginning with “since”

Response 2: We have edited this sentence. The revised manuscript reads: “On the other hand, since the regulation of integrin ……” (lines 83 in the revised manuscript).

Point 3: Line 80. “a priori” should be in italics. 

Response 3: We have edited this sentence. The revised manuscript reads: “…., a priori, chondrocytes can potentially……” (lines 85 in the revised manuscript).

Point 4: Line 134. “5μm” a space is needed here. 

Response 4: We have edited this sentence. The revised manuscript reads: “Both right and left knee joints were serial coronal sections (5 μm) and stained by……” (lines 145 in the revised manuscript).

Point 5: Line 188. “indicate elevated expression” maybe better “indicated”? Moreover, I would re-consider the use of “elevated” and I would consider its substitution. 

Response 5: We have edited this sentence. The revised manuscript reads: “The results of the fluorescence signal indicated elevated expression of aggrecan fragments, MMP13, ……” (lines 222 in the revised manuscript).

Point 6: Line 191. “we also detected” generally I do not advice the use of personal expressions such as “we”. 

Response 6: We have edited this sentence. The revised manuscript reads: “Additionally, it also detected significant changes in these typified OA ……” (lines 225 in the revised manuscript).

Point 7: Figure 1bcd. Figure 1b seems to be characterized by a composition of two sub-figures. Since the other figures are named as b, c, d, I suggest to split figure 1b in figure b, 1c, and then 1d and 1e. Moreover, these figures have not comparable x and y scales. I suggest to use the same scale for axis. 

Response 7: We appreciate the reviewer’s comments. We have edited Figure 1b–e and their scale bars; please refer to the revised version (lines 197–213 in the revised manuscript).

Point 8: Line 242. It is not necessary to add this line reporting “Figure 1”. Same observation for figure 2 and figure 3, that are even included in the diagrams areas. Please remove them. 

Response 8: We appreciate the reviewer’s comments. We have edited all figures; please refer to the revised version.

Point 9: Figure 3c are not readable histograms. Please, enlarge them.

Response 9: We appreciate the reviewer’s comments. We have enlarged Figure 3c; please refer to the revised version. An updated version of Fig 3c is also presented below.

Round 2

Reviewer 3 Report

1. The manuscript title needs to modify

2. Represented cartilage images did not match the OARSI grade. The review could not find any over 1 OARI grade in the presented images. 

3. Due to GRGDS data, it is necessary to modify the title. 

Author Response

Response to Reviewer 3 Comments

Point 1: The manuscript title needs to modify

Response 1: We appreciate the reviewer’s comment. We have changed the title to "Characterization and advancement of an evaluation method for the treatment of spontaneous osteoarthritis in STR/ort mice: GRGDS peptides as a potential treatment for osteoarthritis ", please refer to the revised version.

Point 2: Represented cartilage images did not match the OARSI grade. The review could not find any over 1 OARSI grade in the presented images.

Response 2: We appreciate the reviewer’s comment, we also apologize for our unclear description. Please allow us to add the following points. First, the scoring method of OARSI has been described in the method section of the revised version and the reference paper is stated. Please refer to the “method section” in the revised version, line 143. And also refer to the references section, which is “23. Glasson SS, Chambers MG, Van Den Berg WB, Little CB. The OARSI histopathology initiative - recommendations for histological assessments of osteoarthritis in the mouse. Osteoarthritis Cartilage 2010; 18 Suppl 3:S17-23”, line 531.

Second, the literature is as follows:

Third, in my study, according to our observation, a smooth cartilage surface and normal morphology of chondrocyte nucleus were observed in 6- and 10-week-old mice (Fig. 1a). The fibrillar cartilage was observed in 14-week-old mice, with an average score of 1. The cartilage surface showed vertical clefts and decreased the number of chondrocytes observed in 22-week-old mice, with an average score of 3. Moreover, cartilage morphology was significantly lost in 40-week-old mice, with a decreased and disorganization of chondrocyte columns in the growth plate with age, with an average score of 4 (Fig. 1d-e and Supplemental Fig. 1b).

Point 3: Due to GRGDS data, it is necessary to modify the title.

Response 3: We appreciate the reviewer’s comment. We have changed the title to "Characterization and advancement of an evaluation method for the treatment of spontaneous osteoarthritis in STR/ort mice: GRGDS peptides as a potential treatment for osteoarthritis ", please refer to the revised version.

Round 3

Reviewer 3 Report

Please check the standard OARSI grade, it will be better use other modified OARSI grade.